# Fetal Lung Interstitial Tumor (FLIT): Review of The Literature

**DOI:** 10.3390/children10050828

**Published:** 2023-05-02

**Authors:** Silvia Perin, Ivana Cataldo, Francesca Baciorri, Luisa Santoro, Angelo Paolo Dei Tos, Maria Guido, Paola Midrio

**Affiliations:** 1Pediatric Surgery Unit, Cà Foncello Hospital, 31100 Treviso, Italy; 2Pediatric Surgery Unit, Department of Women and Child Health, University of Padua, 35141 Padova, Italy; 3Pathology Unit, Cà Foncello Hospital, 31100 Treviso, Italy; 4Department of Pathology, Azienda Ospedaliera Università Padova, 35141 Padova, Italy

**Keywords:** FLIT, lung tumors, congenital pulmonary malformations, respiratory distress

## Abstract

Fetal lung interstitial tumor (FLIT) is an extremely rare pediatric lung tumor that shares radiological features with congenital pulmonary malformations (cPAM) and other lung neoplasms. A review of the literature, together with the first European case, are herein reported. A systematic and manual search of the literature using the keyword “fetal lung interstitial tumor” was conducted on PUBMED, Scopus, and SCIE (Web of Science). Following the PRISMA guidelines, 12 articles were retrieved which describe a total of 21 cases of FLIT, and a new European case is presented. A prenatal diagnosis was reported in only 3 out of 22 (13%) cases. The mean age at surgery was 31 days of life (1–150); a lobectomy was performed in most of the cases. No complications or recurrence of disease were reported at a mean follow-up of 49 months. FLIT is rarely diagnosed during pregnancy, may present at birth with different levels of respiratory distress, and requires prompt surgical resection. Histology and immunohistochemistry allow for the differentiation of FLIT from cPAM and other lung tumors with poor prognosis, such as pleuropulmonary blastoma, congenital peri-bronchial myofibroblastic tumor, inflammatory myofibroblastic tumor, and congenital or infantile fibrosarcoma.

## 1. Introduction

Neonatal lung masses are considered rare diseases. They are thought to affect approximately 1 in 15,000 live births (even if some think they might be higher [1]) and are mainly caused by the spectrum of congenital pulmonary airway malformations (cPAM) and, rarely, by the primary lung tumors [2]. cPAMs are mostly benign, but malignancy is, nowadays, a well-known event that can occur throughout life [3]. The differential diagnosis among these entities is crucial and requires histological confirmation, as imaging alone may be inconclusive.

Among the most well-known perinatal tumors, such as pleuropulmonary blastoma (PPB), congenital peri-bronchial myofibroblastic tumor (CPMT), inflammatory myofibroblastic tumor (IMT), congenital or infantile fibrosarcoma [4,5], fetal lung interstitial tumor (FLIT) represents a relatively new entity, which is so rare that has not entered yet in the World Health Organization classification of lung tumors [6]. It was firstly described in 2010 by Dishop [7] in a case series composed of 10 infants, that showed symptoms ranging from the prenatal period to 3 months. At the histological level, the lesions were single solid or mixed solid/cystic lung masses constituted by immature interstitial mesenchymal tissue associated with irregular air space-like structures, mimicking an abnormal fetal lung [7]. A review of the literature, including the first European case from our institution, is herein reported.

## 2. Materials and Methods

A systematic review of the literature reporting cases of FLIT was conducted following the 2020 Preferred Reporting Items for Systematic Review and Meta-analyses (PRISMA) guidelines [8]. The keyword “fetal lung interstitial tumor” was searched independently by two researchers (PM and SP) on the online databases PUBMED, Scopus, and SCIE (Web of Science) until 14 April 2023. Additional reports identified through manual website or citation chain searches were included. Only case reports, original articles, letters, and reviews describing cases of FLIT in the English language were included. Exclusion criteria were non availability of full text, and reports in languages other than English. Duplicates based on author names, titles, and year of publication were removed using Endnote X20 (Thomson Reuters, Toronto, ON, Canada) or manually. All cases of FLIT identified within this search were included in this systematic review. 

Gender, age at presentation, symptoms, age at surgery, tumor localization, type of surgical resection, tumor size, gross features, and recurrences are reported. Statistical analyses were conducted with the R Library ggstatsplot [9].

## 3. Results

The systematic search yielded a total of 77 records published until April 2023: 15 from PubMed, 47 from Scopus, and 15 from SCIE (Web of Science). A flow chart of the number of included and excluded studies for each step of the 2020 PRISMA guidelines is shown in Figure 1. After automated and manual removal of duplicates, the remaining 53 records were screened for relevant titles and abstracts. At this step, abstracts not available in the English language were considered not eligible for future analyses. A total of 40 records were excluded. We then searched for full texts of the remaining 13 records and of these, 12 full texts were available and therefore reviewed. After reading the 12 reports, two were discarded: one for the initial misdiagnosis of FLIT, not confirmed at histology, and the other for the absence of description of a clinical case of FLIT. Ten full-text reports were analyzed. Moreover, 1 additional report, leading to a total of 11 reports describing 21 cases of FLIT, was added after the website and citation chain searches. In particular, one recently published record [10] was retrieved from manual search on websites. By citation chain search, we also identified four additional FLIT cases. Unfortunately, we were not able to include these cases as three reports were not written in the English language (and among them, two were duplicates of records retrieved during the databases search), and one record was not retrieved (Figure 1).

Overall, 12 articles have been included which describe a total of 21 cases of FLIT and its related diagnosis and treatment. All cases of FLIT are summarized in Table 1, including the last one, later described in this work. A prenatal diagnosis was reported in only 3 cases out of 22 (13%). Symptoms, that were mainly represented by respiratory distress, were present at birth in 15/22 cases (68%) and, overall, mean age at presentation was 8 days (with a range spanning from 0 to 90 days of life). A significant prevalence of FLIT in males was evident (14 males versus 7 females and 1 unspecified case, Figure 2A: χ²(2) = 11.55, *p*-value < 0.05). On the other hand, there was no prevalence between right or left lung involvement (Figure 2B: χ²(1) = 0.73, *p*-value = 0.39) as well as among lobes involved (Figure 2C: χ²(5) = 6.36, *p*-value = 0.27). The mean age at surgery was 31 days of life (ranging from the first day of life to 150 days; Figure 2D). A lobectomy was the treatment of choice in 17 cases (77%), while wedge resection was performed in the remaining cases. In one case, the type of resection was not specified, and in another case, the lobectomy was incomplete; however, no recurrence was evidenced at the 3-year follow-up. Tumor size varied between 2 and 9 centimetres with an average of 5.58 (Figure 2E). Overall, no complications or recurrence of disease were reported at a mean follow-up of 4 years, with a maximum follow-up of 15 years (Figure 2F).

The first European case of FLIT is herein described in detail. A full-term male neonate, with no prenatal history, presented persisting polypnea and desaturation a few hours after birth. A thoracic X-ray showed the presence of a round-shaped solid mass of 4 × 3.5 cm in the lower right lobe (Figure 3A). Tumor markers, such as α-fetoprotein, β-HCG, and NSE, were within normal range for the age of the patient. MRI and CT-scans confirmed the presence of a lobulated oval solid mass located in the apical segment of the lower right lobe (Figure 3B,C). The patient underwent a right lobectomy at 18 days of life. Intraoperatively, a brownish solid nodule of 3.5 × 3 cm surrounded by lung parenchyma was detected. At gross examination, the sample was well demarcated and extensive sampling for histological analysis was performed. At histology, the presence of mesenchymal spindle cell proliferation, expanding the pulmonary interstitium, was described. The cells showed a monotonous appearance, characterized by a regular ovoid nucleus and clear-to-pale pink cytoplasm, lacking significant cytological atypia, high mitotic index, and necrosis. Interspersed ectatic thick vessels and scattered spindle myofibroblast-like cells with focal glandular structures resembling immature fetal bronchioles were also detected. Finally, isolated inflammatory cells were also identified. At immunohistochemistry, the stained mesenchymal cells were positive for vimentin and CD99 while they were negative for desmin, myogenin, CD34, and smooth muscle actin (SMA) (Figure 4). Fluorescence in situ hybridization revealed the absence of trisomy 8, 2, and molecular targeted sequencing excluded the presence of a DICER1 mutation. All the morphological, immunohistochemical, and molecular characteristics suggested the diagnosis of FLIT [11,12]. At 14 months from surgery, the patient remains asymptomatic, without evidence of recurrences.

**Table 1 children-10-00828-t001:** FLIT review cases.

Author	Case Number	Gender	Age at Presentation	Symptoms at Presentation	Age at Surgery	Tumor Location	Treatment	Tumor Size (cm)	Gross Features	Outcome
Dishop et al., 2010 [7]	1	M	1 day	-	3 days	Left upper lobe	Lobectomy	3.3	Tan–pink–yellow friable mass	No recurrence, 15 months
	2	M	Birth	-	5 months	Right upper lobe	Lobectomy	4	-	No recurrence, 35 months
	3	F	Birth	-	5 days	Right lower lobe	Wedge resection and subsequent lobectomy	-	-	No recurrence, 32 months
	4	M	Birth	Hydrops and fetal heart failure	Birth	Right lower lobe	Lobectomy during EXIT procedure	5.7	Solid; soft spongy hemorrhagic mass	No recurrence, 60 months
	5	M	3 months	-	5 months	Right lower lobe	Wedge resection	2	Solid; well-circumscribed tan fleshy mass	No recurrence, 36 months
	6	F	2 months	-	2 months	Right lower lobe	Lobectomy	-	Solid; well-demarcated gray–white homogeneous nodule in peripheral lung	No recurrence, 180 months
	7	F	5 days	-	6 weeks	Left lower lobe	Lobectomy	6.6	Solid; soft spongy yellow mass well-circumscribed, distinct from adjacent lung	No recurrence, 19 months
	8	M	2 days	-	11 weeks	Right middle lobe	Lobectomy	4	Tan–pink, spongy, and cystic mass	No recurrence, 23 months
	9	M	Birth	-	2 days	Left lower lobe	Wedge resection	7	Reddish–tan spongy mass	No recurrence, 9 months
	10	M	Birth	-	7 days	Major fissure of the right lung	Wedge resection	6.5	Tan spongy mass	No recurrency at time of publication
de Chadarévian et al., 2011 [13]	11	F	Prenatal diagnosis	-	Newborn	Left lower lobe	Lobectomy	-	Soft; gray–tan, well-circumscribed, surrounded by a fibrous capsule; large cystic space ranging from 0.1 to 1 cm	No recurrence, 7 years
Lazar et al., 2011 [14]	4	M	See case 4		-	-	-	-	-	No recurrence, 60 months
Yoshida et al., 2013 [15]	12	F	7 days	Respiratory distress	13 days	Left lower lobe	Lobectomy	5	Solid; well-circumscribed, surrounded by a thick fibrous capsule	No recurrence, 15 years
Onoda et al., 2014 [16]	13	M	Birth	Mild apnea	11 days	Left lower lobe	Wedge resection	2.5	Solid; well-circumscribed, spongy mass with microcystic component	No recurrence, 36 months
Waelti et al., 2017 [4]	14	M	Birth	Respiratory distress	-	Left upper lobe	Lobectomy, but incomplete resection	8.5	Solid; with microcystic component	No recurrence, 36 months
	15	M	Birth (prenatal diagnosis at 34 weeks of gestation)	Severe respiratory distress	-	Right upper lobe	Lobectomy	9	Solid	No recurrence at first control
Phillips et al., 2019 [17]	16	-	Birth	Severe respiratory distress	20 days	Left upper lobe	Lobectomy	6	Spongy, with cystic spaces	-
Shah et al., 2021 [18]	17	M	Birth	-	21 days	Left upper lobe	Lobectomy	6.5	Solid; soft lobulated with cystic spaces with thick fibrous capsule	No recurrence, 12 months
Kuroda et al., 2021 [19]	18	M	Birth	Mild respiratory distress	22 days	Right upper lobe	Lobectomy	8.5	Solid; spongy with microcysts and a fibrous capsule	No recurrence, 12 months
Wang et al., 2023 [20]	19	M	Birth (prenatal diagnosis at 33 weeks of gestation)	Tachypnoea and feeding difficulties	14 days	Right upper lobe	Lobectomy	6.2	Spongy, well-circumscribed mass with cystic spaces	No recurrence, 3 months
Ho et al., 2023 [10]	20	F	Birth (prenatal history of polyhydramnios)	Respiratory distress	3 days	Right lower lobe	Lobectomy	-	Encapsulated; enlarged air spaces with widened septa	No recurrence, 120 months
	21	F	Birth	Respiratory distress	3 days	Right upper lobe	Lobectomy	-	Irregular air spaces and widened septa, partially encapsulated	No recurrence, 36 months
Present case	22	M	Birth	Respiratory distress	18 days	Right lower lobe	Lobectomy	3.5	Solid	No recurrence, 14 months

## 4. Discussion

Fetal lung interstitial tumor (FLIT) has been described as a solid, spongy, or microcystic lesion, well demarcated from the lung parenchyma, with or without a fibrous capsule [7]. Histologically, FLIT features include immature airspaces and expanded interstitium that resemble fetal lung tissue at canalicular stage, normally seen between 20 and 24 weeks of gestation [15]. Histological examination allows for the correct diagnosis of FLIT and differential diagnosis with other congenital lung masses, such as the pleuropulmonary blastoma (PPB), congenital peri-bronchial myofibroblastic tumor (CPMT), inflammatory myofibroblastic tumor (IMT), and congenital or infantile fibrosarcoma [4,5].

Interestingly, the presentation of FLIT is mainly characterized by respiratory distress at birth, with only a few cases of prenatal signs, such as fetal hydrops, mediastinal shift, and heart failure. Indeed, most fetal lung masses are asymptomatic, with only a small percentage of them growing faster than the fetal chest, therefore causing symptoms [2]. In these latter cases, the induced thoracic compression can result in the development of hydrops as reported by Lazar [14]. However, while cPAMs have been shown to reach the volume growth peak between 20 and 26 weeks of gestation, thus making it feasible to recognize them on routine ultrasound at 20–22 weeks [21], the neoplastic lesions seem to grow later in gestation [2], often remaining undetected until after birth [4]. Moreover, cPAMs have shown that once they reach their volume peak, they tend to remain stable, while tumors keep growing, causing thoracic compression. Indeed, a retrospective study conducted by Waelti [4] determined that among 135 congenital lung lesions, the great majority of them were detected at prenatal ultrasound except for four tumors. For this reason, they concluded that all lung masses discovered during infancy without prenatal diagnosis should be considered as potential neoplastic lesions and therefore removed [4].

As previously stated, the differential diagnosis between different types of lung tumors and pulmonary malformations is a crucial aspect in the management of congenital lung masses. PPB is the most common lung tumor in childhood, affecting children younger than 6 years of age [22]. PPB is conventionally divided into three types according to morphologic features, showing increased behavioral aggressiveness, from type I (purely cystic), to type II (solid/cystic), and type III (purely solid) [7]. The purely cystic type I PPB originates at around 30 weeks of gestation from the proliferation of the interstitial mesenchymal cells, giving rise to a cystic lesion that can be detected at the prenatal ultrasound between 31 and 35 gestational weeks [4]. However, it cannot be distinguished from other cystic lesions detected at this stage, in particular cPAM type I. PPB type I is characterized by a multiloculated cyst lined by cuboidal epithelium. The different cysts are separated by fibrous septa that contain a proliferation of undifferentiated small round cells showing a focal rhabdomyoblastic differentiation and diffuse expression of SMA and myogenin. On the other hand, PPB types II and III, in particular the purely solid one, usually appear beyond the first year of life. The solid area in type II and type III PPBs is composed of either blastematous and sarcomatous elements or nodules of cartilage [5]. 

At molecular level, the identification of the DICER1 mutation [12] or presence of trisomy 8 [23] and 2 [24] have been associated with PPB. Removal of the lesion is considered the gold standard for the treatment of PPB; however, post-surgical chemotherapy can be required, especially if a complete resection is not feasible, due to the risk of recurrence of the lesion in a more aggressive form. On the other hand, no recurrences have been described so far after resection of FLIT, even when the resection was not complete [4]. Moreover, FLIT does not require metastatic assessment since no metastases have ever been diagnosed.

Similarly to FLIT, CPMT can also be considered a benign tumor, which has been described as a solid lesion that can be identified during fetal life, as early as 12 weeks of gestation [4]. It has been associated with hydrops fetalis that can lead to intrauterine fetal death or perinatal respiratory distress. At a histological level, it is characterized by the presence of a highly cellular spindle cell proliferation in the interstitium, arranged in fascicles. As in FLIT, the cells show a fibroblastic–myofibroblastic morphology and immunophenotype, although they can show moderate pleomorphism, occasional nucleoli, and high mitotic index. No atypical mitosis is usually detected. Despite all these morphological worrisome features, CPMT is a benign tumor with a good prognosis [7,25].

IMT is one of the most common primary tumors of the lung that can be found in the pediatric age period. When it is symptomatic, it can cause cough, fever, chest pain, and hemoptysis. On imaging, it appears as a lobulated peripheral mass, sometimes with endobronchial or multicentric growth. The more aggressive forms of IMT can show the infiltration of mediastinum or chest wall as well as vascular invasion [5]. IMT is composed of a uniform spindle cell proliferation with myofibroblastic differentiation and low mitotic index, associated with a chronic inflammatory infiltrate consisting of lymphocytes, plasma cells, and eosinophils. Both these components are merged in a myxoid or collagenous stroma. In more than 50% of cases, IMT presents a molecular rearrangement on the anaplastic lymphoma kinase (ALK) locus on chromosome 2p23 [26]. Although it is considered a benign lesion, it shows an intermediate biologic behavior, with potential risk for local recurrence and, rarely, distant metastasis. The treatment of choice, whenever feasible, is radical surgery. No standard of care has been established yet for advanced cases [5,27,28,29].

Finally, congenital or infantile lung fibrosarcoma is a rare malignant tumor that can appear at different times during the pediatric lifespan, mainly before the age of 5. It can arise everywhere in the body, with pulmonary localization being rarely reported. In this case, respiratory distress and other non-specific signs and symptoms such as fever, cough, hemoptysis, fetal anemia, and pneumonia are reported. Moreover, the tumor might secrete insulin, therefore inducing hypoglycemia. Fibrosarcoma is characterized by the presence of a densely cellulated malignant spindle cell proliferation arranged in interlacing fascicles or a herringbone pattern. Necrosis and hemorrhage can be present as well. Despite a variable morphological spectrum, infantile fibrosarcoma belongs to the NTRK3-rearranged family of tumors. Indeed, more than 70% of cases carry a specific chromosomal translocation (i.e., t(12,15) (p13;q25)) determining the ETV6-NTRK3 fusion gene [5,30]. At present, surgery remains the preferred treatment and, due to the high frequency of recurrences, its association with radiotherapy and chemotherapy can help increase the survival rate. 

The differential diagnosis of FLIT does not comprise only neoplastic lesions but also includes the spectrum of congenital malformations such as CPAM, in particular, the solid form. In fact, the so-called congenital cystic adenomatoid malformation type III appears as a solid lesion, without capsule, characterized by the presence of proliferating small gland-like spaces that resemble terminal bronchiole-like structures lined by ciliated epithelium surrounded by a connective loose stroma. At a variance, type I and II CPAM are described as cystic lesions lined by respiratory epithelium and surrounded by compressed lung parenchyma [7,31]. 

The current literature review has shown that FLIT presents during the early post-natal life with respiratory distress, sometimes with the need for oxygen supplementation, in contraposition to non-neoplastic lung lesions. This is in line with the retrospective analyses of Waelti [4], where 75% of patients with lung tumors were symptomatic and only 13% of those with CPAMs presented symptoms at birth. 

In order to correctly diagnose and treat the solid lung masses presenting at birth with respiratory distress and no prenatal diagnosis, surgery within the first month of life is advisable. Moreover, due to the frequent minor morphological differences among the lesions included in the differential diagnosis, analysis by an expert pathologist is recommended along with proper immunophenotypical and molecular analyses. Finally, difficult cases should always be shared with referral centres to obtain a second expert diagnostic opinion to drive the proper clinical and therapeutic management of all these patients.

## Figures and Tables

**Figure 1 children-10-00828-f001:**
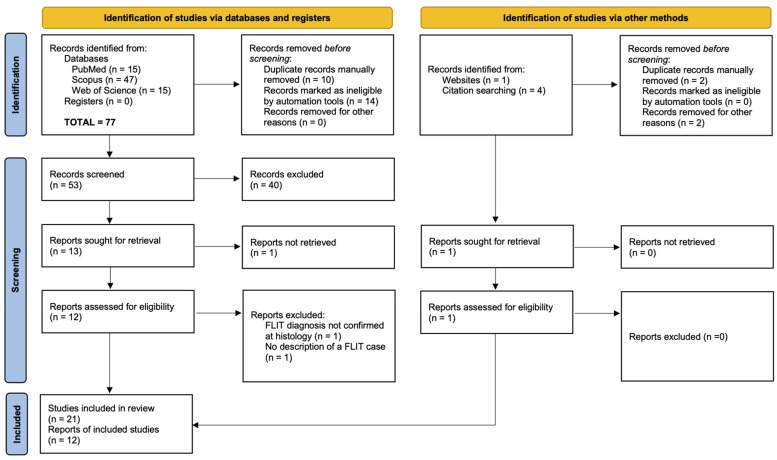
PRISMA 2020 flow diagram for new systematic reviews which included searches of databases, registers, and other sources of “fetal lung interstitial tumor” cases.

**Figure 2 children-10-00828-f002:**
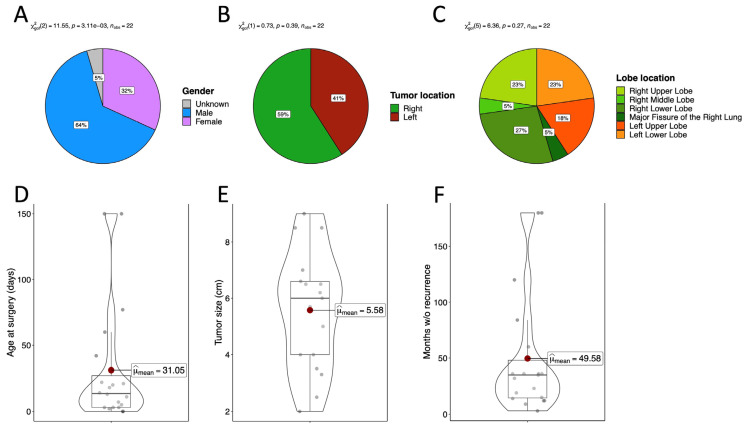
(**A**) Gender distribution. (**B**,**C**) Tumor location. (**D**) Age at surgery. (**E**) Tumor size at resection. (**F**) Months without recurrence.

**Figure 3 children-10-00828-f003:**
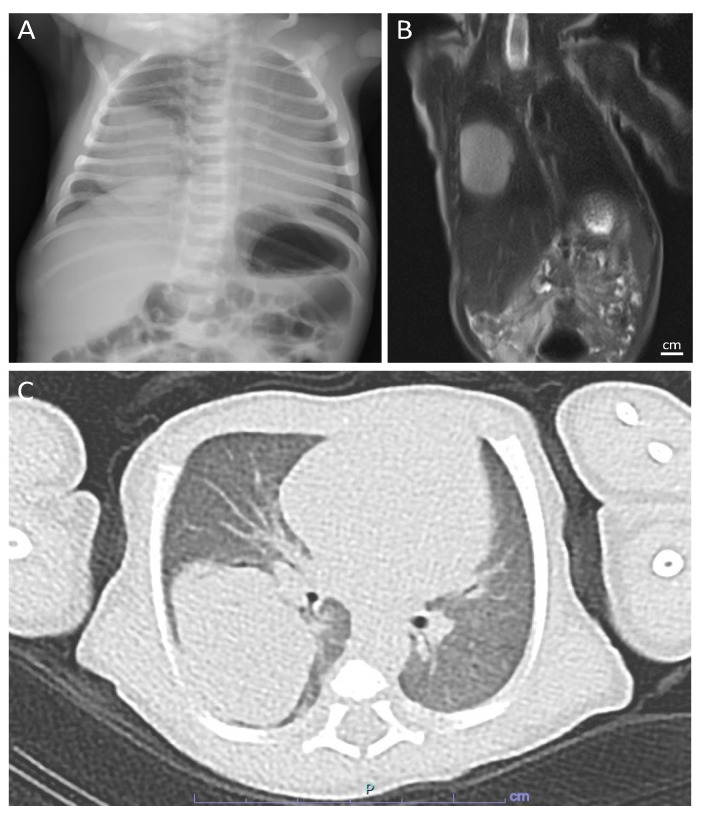
Radiological imaging. (**A**) X-ray image at 2 days of birth. (**B**) MRI in T2 phase showing an oval solid mass located at the lower right lobe. (**C**) TC image confirming the presence of solid mass at the lower right lobe.

**Figure 4 children-10-00828-f004:**
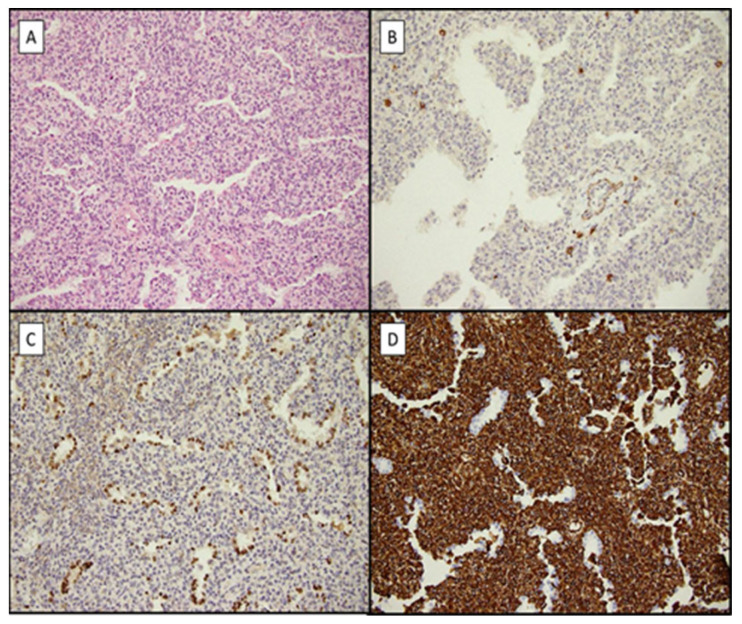
FLIT histological findings. (**A**) FLIT showing bland ovoid interstitial cells with scattered cystic pseudoglandular spaces and thick-walled vessels (hematoxylin eosin ×200). (**B**) Smooth muscle actin shows only rare myofibroblast-like positive cells (immunohistochemistry, ×200). (**C**) Almost every cuboidal epithelial cell lining in the pseudoglandular spaces is stained with thyroid transcription factor-1 (immunohistochemistry, ×200). (**D**) Strong and diffuse vimentin positivity in the interstitial tumor component (immunohistochemistry, ×200).

## Data Availability

The dataset supporting the conclusions of this article is included within the article and its additional file.

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
