# Peer review of "Fetal Lung Interstitial Tumor (FLIT): Review of The Literature"

_children, 2023, doi:10.3390/children10050828_

Round 1

Reviewer 1 Report

The paper “Fetal lung interstitial tumor (FLIT): review of the literature” by Perrin et al describes a case and reviews the literature.

1. The same has been published recently: Case Reports Front Pediatr. 2023 Feb 10;10:1045037. doi: 10.3389/fped.2022.1045037. eCollection 2022. A case report of misdiagnosed fetal lung mass and review of the literature Zongyu Wang 1, Chang Xu 1, Taozhen He 1, Miao Yuan 1

They reviewed the literature on FLIT and found that, to date, 23 cases have been reported worldwide.

The editor needs to decide if this is redundant or not. If publication is recommended, the discrepancy between this study (18 cases in the world literature) and the 23 found in the just published review needs to be clarified. The previous paper should be cited. 

2. How regular and at what exact gestational ages were ultra-sound studies done in this case? This is important information on growth of the tumor. Also indicate what ultrasound chould have detected.

 3. The authors recommend “cases should always be shared with referral centres to obtain a second expert diagnostic opinion”; where did they send their case and what was the result.

Author Response

Thank you for your review and suggestions.

  • The same has been published recently: Case Reports Front Pediatr. 2023 Feb 10;10:1045037. doi: 10.3389/fped.2022.1045037. eCollection 2022. A case report of misdiagnosed fetal lung mass and review of the literature Zongyu Wang 1, Chang Xu 1, Taozhen He 1, Miao Yuan 1

Thank you for the note. The paper has been added to the reference list and included in the review and discussion of the present work.

The discrepancy between our study (18 cases retrieved from the English literature before February 2023) and the 23 found in the just published review, is due to the fact that our study includes papers published in the English literature only. In the above mentioned paper there are 3 cases published in Chinese language, therefore not suitable for our review (inclusion and exclusion criteria have been specified in the text). For the other 2 cases, we could retrieve only the case of ref n.17. However, it turned out to be an abstract presented to a Conference, therefore not a peer-reviewed paper and for this reason it does not appear in any medical database. Finally, the case of ref n.2 (Mocayar Marón FJ, Oliva J, D’Angelo CR, Drago G, Sarabia E, Herón A, et al. Fetal lung interstitial tumor (FLIT): case report. Biocell. (2019) 43(Supplement 4) does not appear in any accessible database, therefore it is not included in our review. 

Finally, while re-running the search of the literature to include the above-mentioned paper, we were able to retrieve another very recent publication on the first 2 Australian cases (Kah Ann Ho et al.  Fetal Lung Interstitial Tumor (FLIT): A case series. Journal of Neonatal Surgery | Year: 2023 | Volume: 12). This reference was not yet available when our search was performed, but now it has been added to the reference list and included in the review.

We hope the implementation of our work will satisfy the referee expectation and will help shearing the knowledge in the field of this very rare disease.

  • How regular and at what exact gestational ages were ultra-sound studies done in this case?

Thank you for the question.

Actually, an ultrasound was performed every trimester (according to national guidelines for uncomplicated pregnancies. All examinations were performed at another hospital). In particular, by reviewing the images taken at 31 weeks’ gestation, no lung masses were visible on the available images and no further examinations were performed. It may be possible the mass was extremely small at this stage, with an echogenic appearance similar to the normal lung tissue, and it was overlooked. Indeed, being considered as a normal pregnancy, the patient was born in a country-side hospital and transferred to our referral center only when symptoms and signs of lung mass were detected.

  • The authors recommend “cases should always be shared with referral centres to obtain a second expert diagnostic opinion”; where did they send their case and what was the result.

Thank you for the question.

Indeed, this case was discussed on the online pediatric oncologic regional platform as soon as the radiologic images (CT and MRI) were performed and, once agreed on the treatment, lobectomy was performed at our center. The oncologic follow-up is currently ongoing at our center and periodically shared within the pediatric oncologic regional platform.

Finally, the manuscript was entirely reviewed to address possible minor spell corrections.

Reviewer 2 Report

The authors present an interesting review on Fetal lung interstitial tumor (FLIT) together with their own case report. FLIT is a true rare disease and therefore this review could be useful and educational. 

The case is well described and sound.

Nevertheless the review methods reveal several drawbacks. The authors need to perform a proper literature respecting the PRISMA guidelines. Clear exclusion criteria must be provided. 

For the results section it is absolutely necessary to provide a flow-chart on the number of articles screened, excluded for various reasons and finally included in the study.

Author Response

Thank you for your review and suggestions.

  • The authors present an interesting review on Fetal lung interstitial tumor (FLIT) together with their own case report. FLIT is a true rare disease and therefore this review could be useful and educational. 

The case is well described and sound.

Thank you for your kind comment.

Nevertheless the review methods reveal several drawbacks. The authors need to perform a proper literature respecting the PRISMA guidelines. Clear exclusion criteria must be provided. 

For the results section it is absolutely necessary to provide a flow-chart on the number of articles screened, excluded for various reasons and finally included in the study.

Thank you for the suggestion.

Indeed, we have re-run the search according to the PRISMA guidelines for systematic reviews that includes searches of databases, registers and other sources, providing description in the text of inclusion and exclusion criteria (from line 66 to line 97). We take the opportunity to underline that, due to recent publications, we have increased the number of cases of FLIT published in the English literature to 22 (including our). A descriptive flow-chart has been provided (new Fig. 1) and the additional cases have been provided in Tab 1. The new references have been included and analyses re-run. Subsequent results have been modified in the result section of the text (line  118 – 187) and in the new Fig. 2.

We hope these additions to the work will meet the referee expectation.

Round 2

Reviewer 1 Report

Well corrected.

Author Response

Thank you for your kind comment.

Reviewer 2 Report

The authors have well responded to the reviewers comments. The manuscript has improved further.

Author Response

Thank you for your kind comment. With regards to spell check, the manuscript has been further revised.